# Modelling Small-Scale Storage Interventions in Semi-Arid India at the Basin Scale

Robyn Horan [1,*], Pawan S. Wable [2], Veena Srinivasan [3], Helen E. Baron [1], Virginie J. D. Keller [1], Kaushal K. Garg [2], Nathan Rickards [1], Mike Simpson [4], Helen A. Houghton-Carr [1] and H. Gwyn Rees [1]

1   UK Centre for Ecology & Hydrology, Wallingford, Oxfordshire OX10 8BB, UK; heron@ceh.ac.uk (H.E.B.); vke@ceh.ac.uk (V.J.D.K.); natric@ceh.ac.uk (N.R.); hahc@ceh.ac.uk (H.A.H.-C.); hgrees@ceh.ac.uk (H.G.R.)
2   International Crops Research Institute for the Semi-Arid Tropics, Patancheru, Hyderabad, Telangana 502324, India; pawan.wable@gmail.com (P.S.W.); K.GARG@cgiar.org (K.K.G.)
3   Ashoka Trust for Research in Ecology and the Environment (ATREE), Bangalore, Karnataka 560064, India; veena.srinivasan@atree.org
4   HR Wallingford Ltd., Wallingford, Oxfordshire OX10 8BA, UK; M.Simpson@hrwallingford.com
*   Correspondence: rhoran@ceh.ac.uk

**Abstract:** There has been renewed interest in the performance, functionality, and sustainability of traditional small-scale storage interventions (check dams, farm bunds and tanks) used within semi-arid regions for the improvement of local water security and landscape preservation. The Central Groundwater Board of India is encouraging the construction of such interventions for the alleviation of water scarcity and to improve groundwater recharge. It is important for water resource management to understand the hydrological effect of these interventions at the basin scale. The quantification of small-scale interventions in hydrological modelling is often neglected, especially in large-scale modelling activities, as data availability is low and their hydrological functioning is uncertain. A version of the Global Water Availability Assessment (GWAVA) water resources model was developed to assess the impact of interventions on the water balance of the Cauvery Basin and two smaller sub-catchments. Model results demonstrate that farm bunds appear to have a negligible effect on the average annual simulated streamflow at the outlets of the two sub-catchments and the basin, whereas tanks and check dams have a more significant and time varying effect. The open water surface of the interventions contributed to an increase in evaporation losses across the catchment. The change in simulated groundwater storage with the inclusion of interventions was not as significant as catchment-scale literature and field studies suggest. The model adaption used in this study provides a step-change in the conceptualisation and quantification of the consequences of small-scale storage interventions in large- or basin-scale hydrological models.

**Keywords:** semi-arid hydrology; small-scale storage; check dams; tanks; farm bunds; Cauvery; GWAVA

## 1. Introduction

Water resources management is becoming increasingly challenging [1] with rapid population growth [2], a changing climate [3], and increasing competition over limited natural resources [4]. For centuries, local communities and municipalities have altered the landscape and built informal structures to increase local water security. In semi-arid regions of the world, people have relied on large-scale infrastructures, such as dams and water transfer schemes, and small-scale infrastructures, such as check dams, farm bunds (rainfall harvesting method used in agriculture fields consisting of a raised soil perimeter), and tanks (small informal reservoirs with a catchment area of less than 34 hectares), to provide and store water for urban and rural use. Detailed descriptions of these structures can be found in Section 2.2.2.

In India, the shortfall in renewable water resources to meet the increasing demand has resulted in aggressive abstraction of the deep groundwater storage and the construction of

small surface-water storage structures [5]. The Government of India and State governments have actively encouraged the construction of interventions, such as check dams, farm bunds, and tanks, as the primary policy response for alleviating water scarcity [6]. There are now millions of such structures across India [7] and, recently, there has been renewed interest in their effectiveness for improving local water security. It is of critical importance to understand the hydrological effect of these interventions at the local- and basin-scale to inform sustainable water resource management.

In India and other semi-arid regions, interventions are generally constructed to assist in the replenishment and maintenance of local groundwater resources [8]. The most prolific types of interventions in Southern India are check dams, farm bunds, and tanks [9]. In mountainous regions of the world, such interventions are commonly used to reduce the velocity of streamflow and reduce the sediment loss from the catchment. However, in India, these interventions are primarily used for the purpose of artificial groundwater recharge [5]. There is limited knowledge of the hydrological dynamics and performance of interventions [10], and little research has been undertaken to quantify the hydrological effects of interventions at a basin-scale [11]. Some studies have modelled the local impact of interventions on streamflow with different perspectives, including: the impact on the water balance [10], as a possible use to treat wastewater [12], and the impact on river flows in headwater catchments [13,14]. Additionally, many studies have focused on the effects of interventions on sediment transport and local groundwater level [15–23]. The upscaling of small-scale storage interventions is of high interest because it is becoming increasingly popular for water resource management and planning approaches to focus on the basin as an entity [24]. A basin-wide approach is important in semi-arid regions and particularly pertinent in closed and closing basins, where water is a scarce commodity and upstream interventions directly affect downstream water availability [24].

There are concerns regarding the effects and functionality of interventions in Peninsular India. The underlying fissured hard-rock geology of Peninsular India differs from the alluvial deposits in Northern India, where most previous studies have been undertaken. Fissured hard-rock has a medium to low permeability and contains aquifers with modest water resources compared to porous, karst, and volcanic aquifers. These aquifer systems are well-connected and have transitioned from a laterally to a vertically dominated flow system due to high levels of abstraction. These aquifers have decreasing hydraulic conductivity and storage with depth, resulting in the resource becoming depleted quickly when low levels of pumping are reached. The aquifer systems are now regarded as either depleted or highly variable resource due to the high levels of abstraction and seasonal recharge. It is speculated that the abstraction from these aquifers is resulting in decreased base flow into the river system [5].

The Cauvery Basin was chosen to be representative of many other basins in Peninsular India. These basins are under pressures of urbanisation, population growth, and agriculture intensification [24]. The Cauvery is additionally a contentious river with concern over sharing of water between Karnataka and Tamil Nadu [25]. With water resources in the Cauvery Basin under severe stress and the abundance of small-scale interventions, it is important to understand the effect of interventions on the spatial and temporal hydrological patterns [11]. There are constraints and uncertainty identified in the current modelling of interventions at the basin scale:

- The hydrological functioning of each type of intervention is uncertain.
- Proxy values and parameter adjustments have been utilised in an attempt to quantify the functioning of interventions.
- Data on the location and characteristics of interventions are scarce and not well documented when available.

The impacts of such changes and interventions on local hydrological processes, such as streamflow, groundwater recharge, and evapotranspiration, are poorly understood, and knowledge of how these diverse local changes cumulatively affect water availability at the broader basin-scale is very limited.

Over recent decades, the hydrological regime of the Cauvery Basin has been significantly altered [26] across the four federal states in which it lies [27]. The basin is highly water-stressed [28] and the current water use exceeds the renewable water resources within the basin. A common technique throughout the four states is the use of small-scale storage structures to assist in the alleviation of local water stress in non-monsoon periods [29]. All the water resources associated with a "normal" rainfall year are currently allocated by tribunal [25], and surface water flows only reach the Bay of Bengal in years of strong monsoons [30]. The agricultural activities across the basin require 90% of the total water resources [31]. However, rapidly developing urban and industrial centres are creating increased inter-sectorial and inter-state competition for limited renewable resources [32]. The four states have different water policies, traditional water harvesting techniques, water use prioritisation, and value associated with the natural environment [33].

Several hydrological modelling exercises have already been carried out in the Cauvery Basin or its sub-catchments. The Auto-regressive moving average time series (ARIMA) model [34], an artificial neuron network (ANN) model [34], support vector regression (SVR) model [34], and the Soil and Water Assessment Tool (SWAT) model [35] have been utilised in various sub-catchments of the Cauvery Basin. At the basin scale, SWAT [36–39], Soil Conservation Service Curve Number (SCS-CN) [40,41], and the coupled mesoscale hydrologic model with the Variable Infiltration Capacity model (VIC-MHM) [42] have been used to simulate streamflow. However, none of these previous studies have considered the inclusion of small-scale interventions.

The use of the GWAVA model in the Cauvery Basin provides the opportunity to investigate the effect of interventions on basin scale hydrology by introducing check dams, farm bunds, and tanks into the model structure. To investigate the effect of the interventions on the hydrology of the Cauvery Basin, a version of the GWAVA model (GWAVA-GW) made specifically for this user was developed. In GWAVA-GW, the groundwater module was modified to better capture groundwater levels. The interventions were conceptualised within the model structure using local knowledge, observed data, and adaptations of existing reservoir representations. The effect of interventions on the hydrological regime and water balance of the entire Cauvery Basin was studied, as well as a more in-depth analysis of two relatively small sub-catchments contained within the basin.

## 2. Materials and Methods

The GWAVA model was used to understand the hydrological functioning and impacts of interventions on the water balance of the Cauvery Basin.

### 2.1. Site Description

The Cauvery River basin is the fourth-largest basin in Peninsular India; it drains an area of 81,155 km$^2$ [43]. The Cauvery originates in the Western Ghats at Talakaveri in the Kodagu district of Karnataka and the head waters of the basin form in the Nilgiri and Anaimalai mountains. The Cauvery Basin is predominantly situated in the federal states of Karnataka and Tamil Nadu, although it crosses into Kerala and Puducherry [27]. The main river channel flows south-easterly through the states of Karnataka and Tamil Nadu to outflow at the Bay of Bengal [44].

The Cauvery Basin is subjected to a large degree of heterogeneity not only in topography and land use, but also in climate and economic development [45]. The landscape is semi-arid with the majority of the basin's water coming from the south-western monsoon in the summer months. The basin experiences distinct intra-annual seasons, namely South-Western (SW) monsoon in the spring, the North-Eastern (NE) monsoon in the autumn, and post-monsoon conditions in the winter. The upper catchment receives rainfall from both the SW and NE monsoons, whereas the lower catchment only receives rainfall from the NE monsoon. The mean annual rainfall varies from 6000 mm in the upper reaches to 300 mm on the eastern boundary [46]. The mean daily temperatures vary between 9 and 25 °C throughout the catchment [26]. The Western Ghats form a rain-shadow along the western

coastline, decreasing the precipitation gradient during the SW monsoon [47]. In addition to difficult water resource management on the ground, high levels of heterogeneity across the basin pose a challenge when undertaking spatial modelling activities, identifying feedbacks and upscaling catchment processes.

The basin is highly anthropogenically influenced. Currently, there are over 100 impounding reservoirs (Figure 1a) and approximately 20 major water transfer schemes within the basin (Figure 1c), along with millions of small-scale interventions throughout the rural and urban regions of the basin [26]. The four major reservoirs that are constructed within the catchment are Kabini (440 MCM), Bhavanisagar (791 MCM), Krishna Raja Sagara (1016 MCM), and Mettur (2640 MCM). The releases from these major reservoirs regulate and disrupt the seasonal trend of the streamflow downstream of their release points. The land use of the basin comprises of 48% agriculture, 22% non-arable land, 19% forest, and 9% urban (Figure 1b) [26]. Natural forests are under great stress due to increasing demand for forest products and competition over land use. Across the basin, approximately 60% of the total population rely on agriculture [26]. Agricultural crops are grown within irrigated canal command areas or in rain-fed areas utilising farm bunding techniques. Paddy and sugarcane, both water-intensive crops, are predominant within the irrigated and rain-fed areas of basin and in the delta regions. The urban areas within the basin have expanded by over 35% over the last decade and are expected to continue to increase with the expanse of industry [48]. With the threat of rising surface temperatures, the competition for fresh water resources between the agricultural sector and other water uses is likely to intensify.

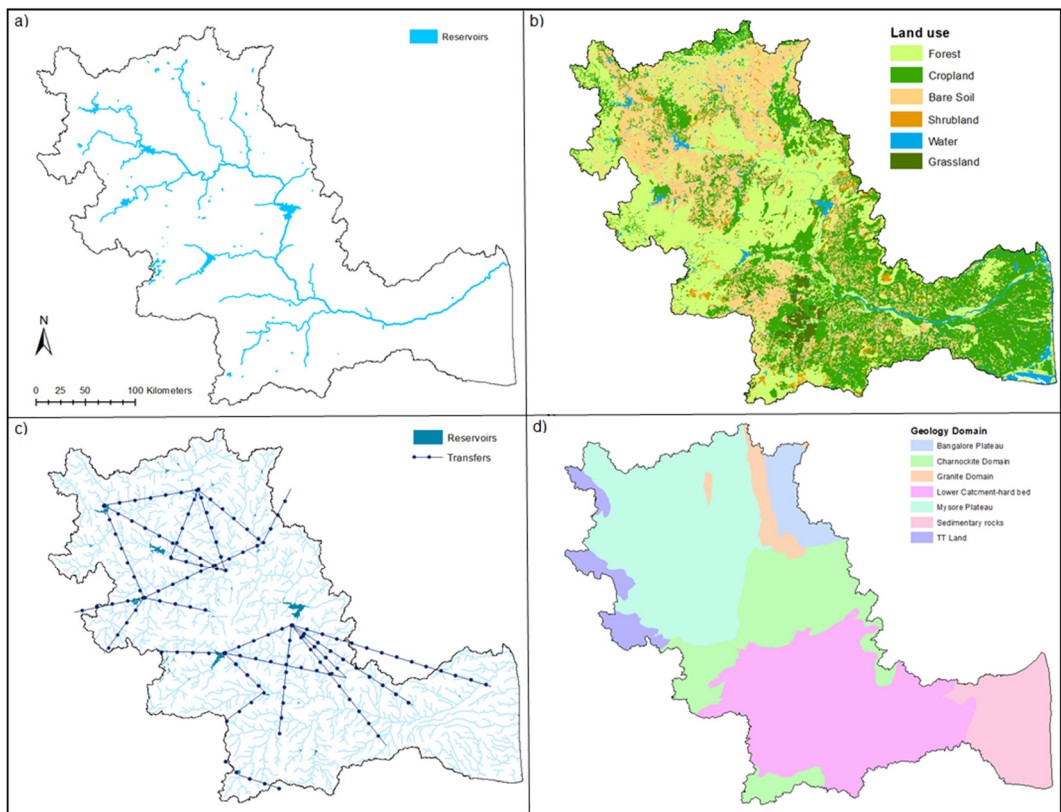

**Figure 1.** Physical characteristics of the Cauvery Basin. Presented are (**a**) the location of reservoirs within the basin, (**b**) the land use of the basin, (**c**) the major reservoirs and the water transfer links constructed in the basin, and (**d**) the geological domains within the basin.

Model-simulated streamflow, total evaporation, water table level, and base flow were investigated at two sub-catchment outlets, located in Karnataka and Tamil Nadu, and the furthest downstream gauging point (Musiri) (Figure 2) to determine the effects of the inter-

ventions on the availability of simulated streamflow and the catchment water balance. The two sub-catchments were selected based on a similar density of interventions but differing underlying geology (Figure 1d). The base flow component (groundwater flowing into the river channel from the aquifer) between the two sub-catchments is therefore different.

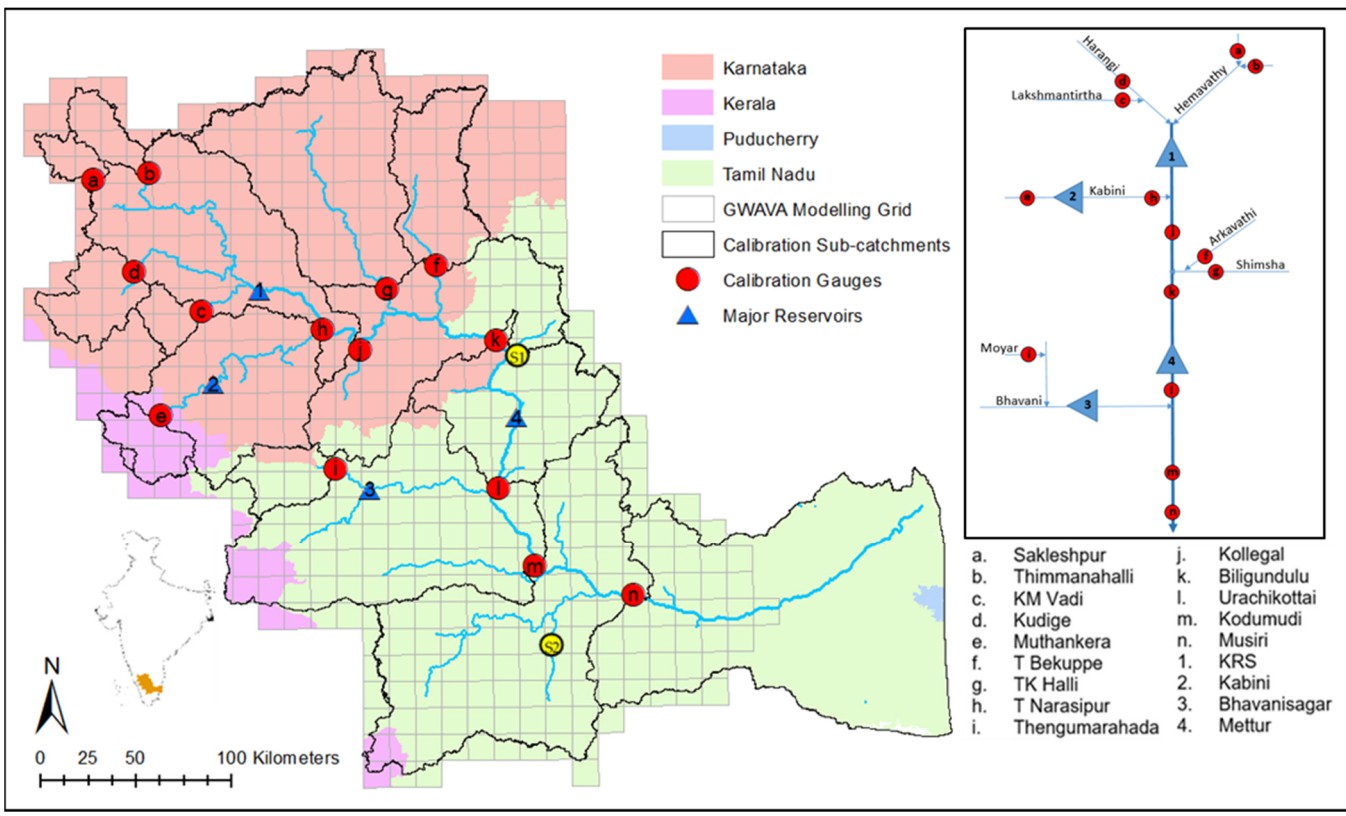

**Figure 2.** Inset: Flow diagram of the Cauvery Basin; main map shows the four states falling within the basin (pastel shading), sub-catchment boundaries, modelling grid, and the locations of the 14 calibration gauges (a–n) and 4 major reservoirs (1–4) within the Cauvery Basin. S1 and S2 (Table 1) represent the points on the stream network where the effect of interventions is investigated.

**Table 1.** The mean annual precipitation (MAP), catchment area (Area), flow characteristics, period of no observed streamflow in main channel ($T_{noflow}$-days of no streamflow), and underlying geology of two sub-catchments used in this study.

| Sub-Catchment Number | MAP (mm) | Area (km$^2$) | Rainfall Period | Flow Characteristics | Period of No Streamflow in Main Channel (Days per Year) |
|---|---|---|---|---|---|
| S1 | 864 | 2660 | March–January | Non-Perennial | $30 < T_{noflow} < 60$ |
| S2 | 867 | 3120 | March–January | Perennial | $0 < T_{noflow} < 3$ |

### 2.2. Model Development

The Global Water Availability Assessment Tool (GWAVA) is a large-scale, semi-distributed gridded water resources model developed by the UK Centre for Ecology & Hydrology [49]. The model incorporates natural processes (soils, land use, lakes, etc.) and anthropogenic influences (crops, domestic and industrial demands, reservoir operations, transfers, etc.). GWAVA estimates local runoff from each cell using a lumped conceptual probability distribution model (PDM) [50]. The PDM requires a limited set of parameters, with the model configuration comprising of three components, namely, the probability- distributed soil moisture storage, the surface storage, and the groundwater storage. GWAVA utilises a combination of land use and soil types. There are four land use options (trees,

shrubs, grass and bare soil) and seven soil type classifications (ranging from sandy to organic). The soil moisture characteristics for each combination are defined by rooting depths, wilting points, field capacities, and saturation capacities. The evaporation is estimated using the FAO-56 Hargreaves equation from both the natural vegetation and agricultural crops, while the effective precipitation is determined using a two-parameter exponential equation, as described by Calder [51]. The soil moisture and direct runoff are calculated separately for each land use type and then summed to obtain a total direct runoff for each grid cell. The total direct runoff is then routed through any existing engineering structures within the grid cell using the Muskingum equation and the user-defined reservoir outflow or transfer parameters. This is followed by a demand-driven routine to account for the anthropogenic stresses on the system. GWAVA accounts for water demands from the domestic (urban and rural), industrial, and agricultural sectors. Domestic, industrial, and livestock demands are user defined and temporally static, but spatially dynamic. Irrigation demand is temporally and spatially dynamic and is estimated via a user-defined crop type and planting month [49]. For this study, two major model developments were undertaken; the first was the inclusion of a demand-driven groundwater routine, and the second, the inclusion of small scale interventions.

### 2.2.1. Groundwater Routine

An improved groundwater module with additional groundwater processes was added to GWAVA to necessitate the full coupling of the water abstractions. The improved groundwater representation is a modified rendition of the AMBHAS-1D model [52]. The groundwater storage for each grid cell comprises of a layered aquifer. The number of layers and the depth of each layer is flexible, and the values for specific yield for each layer are user-defined according to the local hydrogeology. The store is recharged from the soil moisture, lakes and reservoirs, leaking water supply infrastructure, and artificial recharge structures. The recharge to the aquifer from large water bodies is assumed to be at a constant rate specific to each water body, while the recharge from water supply infrastructure is determined by the user-defined recharge fraction of the conveyance loss. The percentage conveyance loss from total demand is set by the user and varies between urban and rural water demands to reflect the different infrastructures. The groundwater storage is routed as base flow using a routing coefficient and a user-defined level of groundwater storage below which there is no base flow. The groundwater storage is converted to an aquifer level (meters below ground level) by dividing by the specific yield. Water can be directly abstracted from the groundwater storage down to the user-defined maximum depth of the aquifer. One limitation of this representation is the lack of lateral flow between groundwater storages. It was decided that neglecting lateral flow was an acceptable approximation given the scale of the model.

### 2.2.2. Conceptualisation of Interventions

The typical characteristics and functioning of each small-scale structure were determined to conceptually represent them in the GWAVA model. Due to the abundance of these small structures throughout the basin, the lack of spatially explicit data and the grid resolution of GWAVA, it was deemed impossible to simulate the effect of each single structure. Instead, each type of intervention was aggregated for every $0.125°$ (approximately 12 km $\times$ 12 km in India) cell to form a single composite tank, check dam, and farm bund within the cell. For this aggregation to be possible, the surface area of each intervention in a cell was required to estimate the total storage capacity for each type of intervention in that cell. The check dams utilised trapezoidal scaling while the tanks and farm bunds utilised cuboidal scaling to determine the storage capacity.

As a result of the structures, the increased open water surface area increases evaporation losses within a grid cell. A constant open water evaporation (OWE) factor was applied to all the interventions. The monthly average OWE was estimated from the evaporation-control-in-reservoirs documentation [53].

- Urban and Rural Tanks

For the purpose of this study, and referring to terminology used in India [12], small, abundant reservoirs with less than 34 hectares of drainage area and built for the decentralised harvesting of runoff, particularly in the monsoon season, are referred to as tanks [54]. These are typically constructed using a shallow dam across a river channel and are unlined [14]. Tanks provide small-scale storage of rainfall and streamflow, control flood waters, and increase recharge to groundwater in the immediate area [55]. Rural tank storage is seasonal [56], and in many semi-arid regions, tanks provide the only means to store rainwater and streamflow for irrigation [57]. Urban tanks are fundamental for city drainage systems [14] used for the collection and recycling of wastewater.

For their conceptualisation within GWAVA, both urban and rural tanks were assumed to have an inflow component comprising of daily rainfall, wastewater, and streamflow within the cell, with spill contributing to the outflow (Figure 3). Furthermore, these tanks are generally unlined in order to help groundwater recharge locally. Thus, a leakage rate of 13 mm d$^{-1}$ [58] and 6 mm d$^{-1}$ [59] was added for the rural and urban tanks, respectively. The recharge from tanks is relatively low as these structures tend to be highly silted and infiltration is limited through the fine particles lining the bottom. The recharge from rural tanks was higher than from urban tanks under the assumption that tanks in rural areas were constructed more recently and dredged more regularly. In the absence of detailed tank bathymetry data, it was assumed that all tanks are cuboid in shape with a maximum depth of 3.0 m deep at full capacity [60].

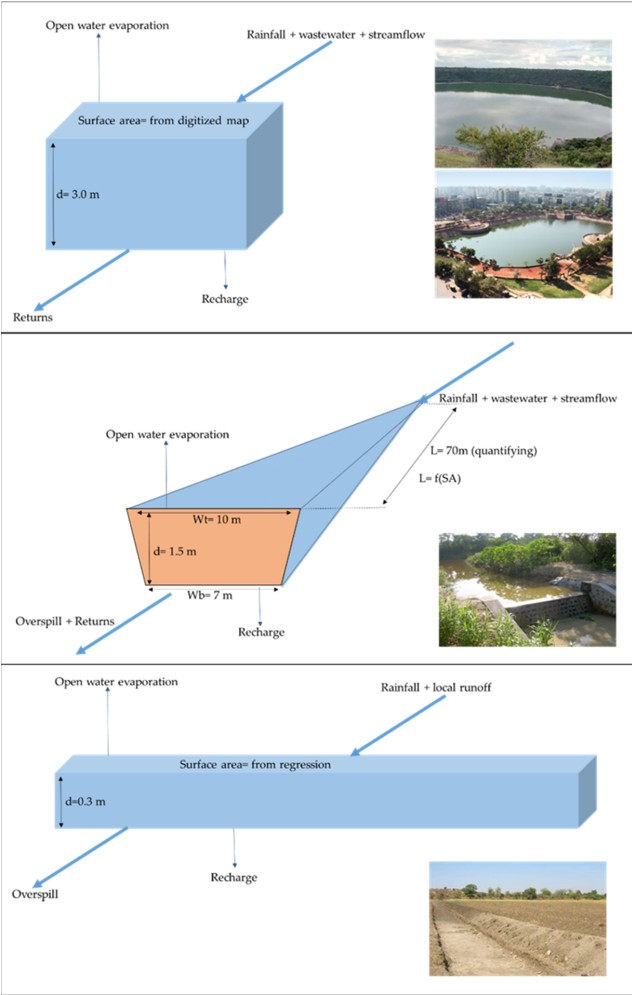

**Figure 3.** Conceptual diagram and a photograph of an example of urban and rural tank (**top**), check dam (**middle**) and farm bund (**bottom**) adopted in the GWAVA model.

- Check Dams

Check dams are small water conservation structures (<0.5 ha) built across a stream using concrete, sandbags, or logs [18]. These are designed to reduce the velocity of streamflow through the catchment and to retain the floodwaters (monsoonal rainfall in the case of India) [11]. The process of impounding water at a local scale is thought to increase the groundwater recharge and soil water potential in the adjoining areas [61].

For the model representation of check dams, it was assumed that daily rainfall, local runoff, and streamflow of the cell contribute to the inflow (Figure 3). From a field study in the Upper Cauvery region, the leakage from the bottom of the structure was assumed to be 100 mm d$^{-1}$ across all the check dams in the catchment [62]. The outflows of the check dams comprise of spills. For the purpose of this study and to simplify data collection of thousands of structures, all check dams in the basin are assumed to have the same dimensions and, thus, capacity. In the absence of comprehensive dimension data from within the basin, assumptions have been made based on available literature. The depth is assumed to be 1.5 m [58], the top width of the structure equal to 10 m, and the channel slope to be 1% [62]. In the absence of data quantifying the number and spatial repartition of check dams in the Cauvery Basin, a surrogate methodology to estimate these alongside with their storage capacity was established. Based on discussions with stakeholders and cited literature [7,63,64], it was assumed that an average check dam in the Cauvery Basin is a 3D trapezoid with a profile that is 10 m in width at a distance of 70 m upstream of the structure. Thus, the surface area of a check dam was assumed to be triangular and fixed at 350 m$^2$ (70 m multiplied by half the wall length) for every check dam included in the model. The assumed average surface area was used solely in the determination of the total surface area of check dams within a grid cell: The number of check dams (see Section 2.2.2) within a cell was multiplied by 350 m$^2$ to determine the surface area of check dams in each cell. Within the model conceptualisation, the length of the conceptual aggregated check dam was dependent on the surface area. The width and depth remained at 10 m and 1.5 m, but the length was variable.

- Farm Bunds

Farm bunding is a traditional in-situ method for soil and water conservation [65]. Bunds are a raised perimeter at the foot slope of agricultural fields, constructed of soil or stone, to increase the time of concentration of precipitation, allowing rainwater to percolate into the soil [66]. Bunds are constructed to retard the movement of overland flow and encourage infiltration within the field [67].

Farm bunds are assumed to be filled from daily rainfall and local runoff within the cell. The saturated hydraulic conductivity of the soils [68] in the basin and the high diurnal temperatures resulted in the water within the farm bunds infiltrating or evaporating completely within a day. The open water evaporation constant was applied to the surface area of the bunds while the infiltration rate differed with regards to soil type. To simulate groundwater recharge from these structures, a rate relative to the saturated hydraulic conductivity of the soil [68] of the area was selected. Once the water held in the bund was at full capacity, excess water could flow over the structure and into the stream. Without adequate field measurements, it was assumed from available literature that all bunds are a maximum of 0.3 m deep [69,70] (Figure 3). The surface area of the farm bunds area is derived in Section 2.2.2.

Following the inclusion of the interventions within GWAVA, a sensitivity analysis, in line with that of Wable et al. (2019) [62], was performed to assess the effect the number and dimensions of the interventions have on the simulated streamflow. A detailed description of the scenarios can be found in Table A1 in Appendix A. For the purpose of this sensitivity analysis, the scenarios presented have been run under natural conditions throughout the catchment to isolate the effects of the dimensions and number of interventions on the streamflow.

### 2.3. Model Application

For this application of the GWAVA model in the Cauvery Basin, a grid cell resolution of 0.125° was chosen based on data availability for the region. The model version including the improved groundwater module and interventions was utillised for this study. The model was set up to include the natural vegetation, agricultural areas, urban areas, rural areas, industrial areas, 5 major reservoirs, 49 minor reservoirs, 27 transfers, and a significant number of interventions. The model was calibrated and validated utilising the availiable uninterupted observed streamflow data of an adequte quality. A baseline period from 1986–2005 was utilised for the analysis presented within this manuscript. Five scenarios were considered to analyse the effects of the interventions within the Cauvery Basin:

1. All interventions (tanks, check dams and farm bunds)
2. No interventions
3. Only tanks
4. Only check dams
5. Only farm bunds

### 2.4. Model Calibration and Validation

GWAVA is calibrated against observed streamflow data using the SIMPLEX auto-calibration routine. This routine uses five parameters for calibration: a surface and ground-water routing parameter, a PDM parameter that describes spatial variation in soil moisture capacity, and a multiplier to adjust rooting depths and level of groundwater storage below which there is no base flow.

GWAVA was calibrated and validated using observed streamflow gauge data from 14 different gauging stations across the basin (Figure 2). The calibration gauges were selected from a set of 28 gauges across the basin based on completeness of the data, time-period of the data, and size of the subcatchment. Data were deemed sufficient when more than 50% of the data points were identified as 'observed' and not 'calculated', and had at least five consecutive years available. However, this threshold may appear low, considering a higher proportion of observed to calculated data left an insufficient number of gauges to choose from. Additionally, subcatchments of fewer than four GWAVA grid cells were excluded. The name of each gauging station and the years used for calibration and validation are presented in Table A2 (Appendix B).

The automatic calibration was run across the 14 delineated subcatchments: It must be noted that the parameters in the auto-calibration routine only affect the natural components of the system. Due to the observed streamflow being highly influenced by the reservoir outflows, a manual calibration was carried out for gauges downstream of reservoirs, by re-running the autocalibration routine with a range of different reservoir parameters.

### 2.5. Data Acquisition

Input data were collected from several sources and extracted from global and regional datasets (Table A3 in Appendix C). Data regarding the number and distribution of interventions in the Cauvery Basin are sparse. Extrapolation and estimation methods described in this section were used to provide the necessary surface area data for input into GWAVA.

The surface areas of the rural and urban tanks were estimated by isolating the 'tanks' from the Cauvery Water Bodies dataset (Figure 4). This dataset consists of a shapefile containing all the medium to large waterbodies (rivers, lakes, reservoirs, tanks, wetlands, etc.) in the Cauvery Basin in 2019, derived using remote sensing techniques. The urban tanks were identified as tanks that fell within urban centre boundaries. The tanks outside of these boundaries were assumed to be rural. Check dams and field bunds are too small to be detected by this methodology.

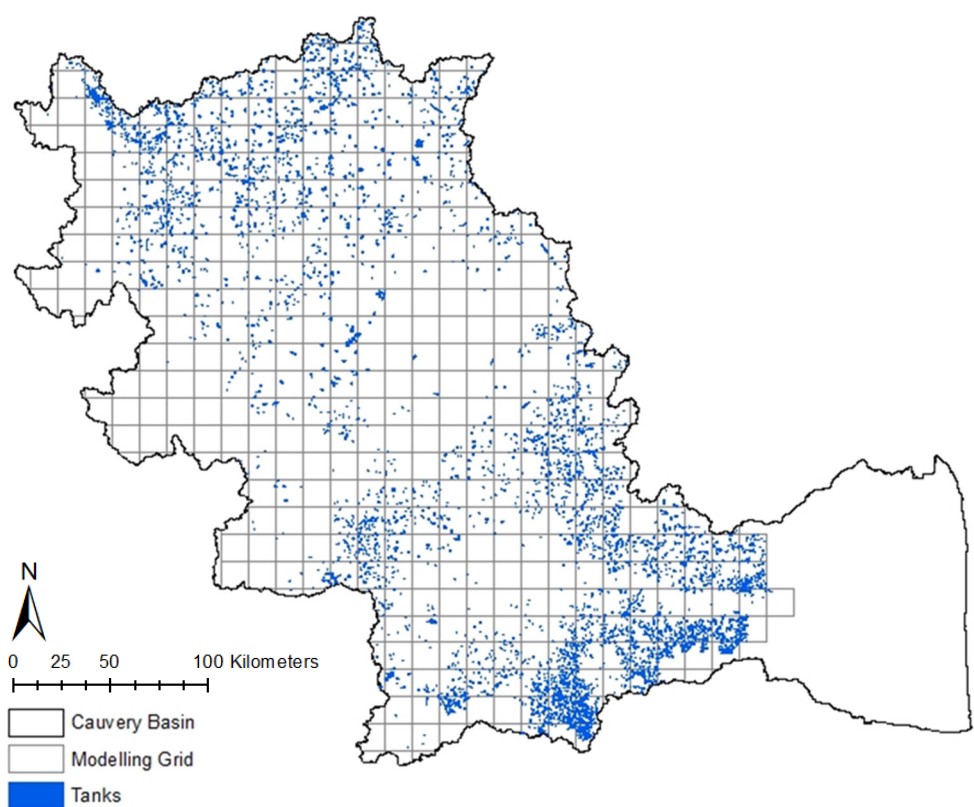

**Figure 4.** The distribution of tanks [71] within the Cauvery Basin superimposed with the modelling grid of 0.125 degree.

Data for the farm bunds and check dams were derived from district-wise Structural Investment Report's available for Karnataka from 2006 to 2012 (Figures 5 and 6). For each district in Karnataka, the area covered by farm bunds and the number of check dams was calculated from this financial data by dividing the total expenditure for each type of intervention by the expenditure per hectare of bunding and of a check dam.

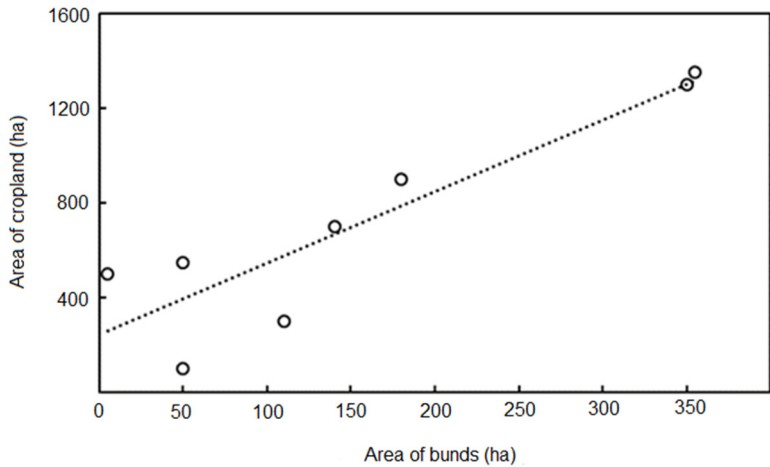

**Figure 5.** Graphical correlation between area of farm bunds (hectares) and area of cropland (hectares) in each district in Karnataka.

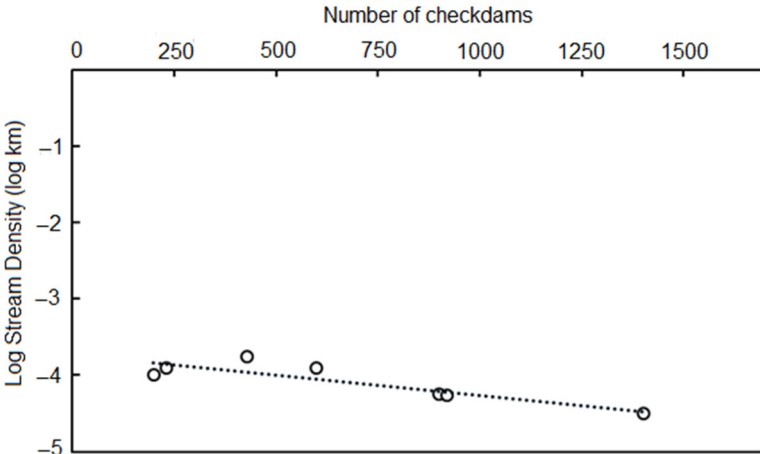

**Figure 6.** Graphical correlation between number of check dams and stream density in each district in Karnataka.

In the absence of data for the state of Tamil Nadu, the data from Karnataka were extrapolated. Plausible relationships between the number of check dams and the area of bunding with soil type, rainfall, slope, population, land type, irrigation type, and geology in a district were all investigated. None of these yielded any significance. Meaningful relationships, however, were drawn between the number of check dams and the stream density, and the area of bunding and the area of rainfed agriculture. These are described below.

Within the districts of Karnataka, a relationship was drawn between the area of farm bunds and the area of rainfed cropland within each district ($r^2$ = 0.91, Figure 5). Due to a lack of data, it had to be assumed that this relationship was also evident in Tamil Nadu.

The regression (Equation (1)) was utilised to estimate the area of farm bunds within each district of Tamil Nadu:

$$A_c = 2.75A_b + 338 \tag{1}$$

where $A_b$ is the area covered by bunding (ha), and $A_c$ is the area of rainfed cropland (ha)

Additionally, a relationship was drawn between the log function of the stream density (SD) of each district in Karnataka and the number of check dams ($r^2$ = 0.93, Figure 6). The stream density is characterised by Equation (2) [72].

$$Log\,(SD) = \sum \frac{Length\ of\ streams\ of\ all\ orders}{Area} \tag{2}$$

As with the farm bunds, it is assumed that this relationship holds true into the districts of Tamil Nadu.

A regression function (Equation (3)) was used to estimate the number of check dams within each district in Tamil Nadu:

$$Log\,(SD) = 0.0017N_{cd} - 4.33 \tag{3}$$

where $SD$ is the stream density, and $N_{cd}$ is the number of check dams.

The district-wise data was applied to the modelling grid using a weighting function of the grid-wise crop area (Figure 7a) and stream density (Figure 7b), respectively. Across the catchment, the surface area of the interventions within each grid cell ranged between 0.02 and 53 km$^2$.

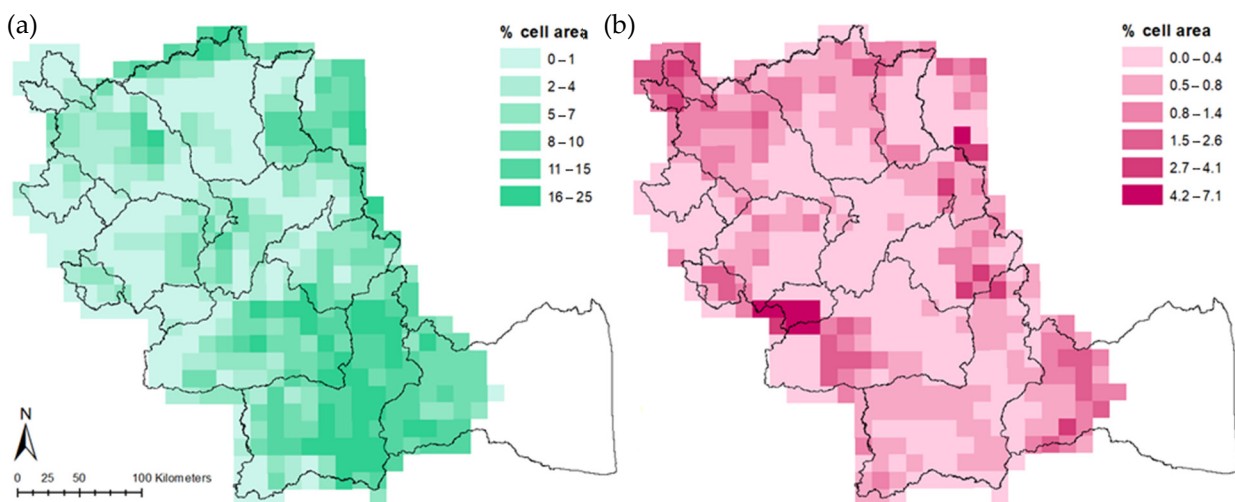

**Figure 7.** The distribution of (**a**) farm bunds and (**b**) check dams as a percentage of the GWAVA modelling grid cell area across the Cauvery Basin.

## 3. Results

### 3.1. Model Performance

The model performed well in the sub-catchments of the upper reaches, but struggled to reliably simulate the flows downstream of the Mettur Dam (Figure 2). Across the calibrated sub-catchments, the model underestimates the total volume of simulated streamflow. The gridded precipitation data [73] produced by the Indian Meteorological Department (IMD) underestimate the point measured rainfall in the region across the Western Ghats by an excess of 50% (Figure 8). This could be the fundamental explanation for the consistent underestimation of simulated streamflow by GWAVA. Inaccurate simulations of the total volume of water within the system and reservoir releases undermine the value of the model's predictive ability as a water resources management tool.

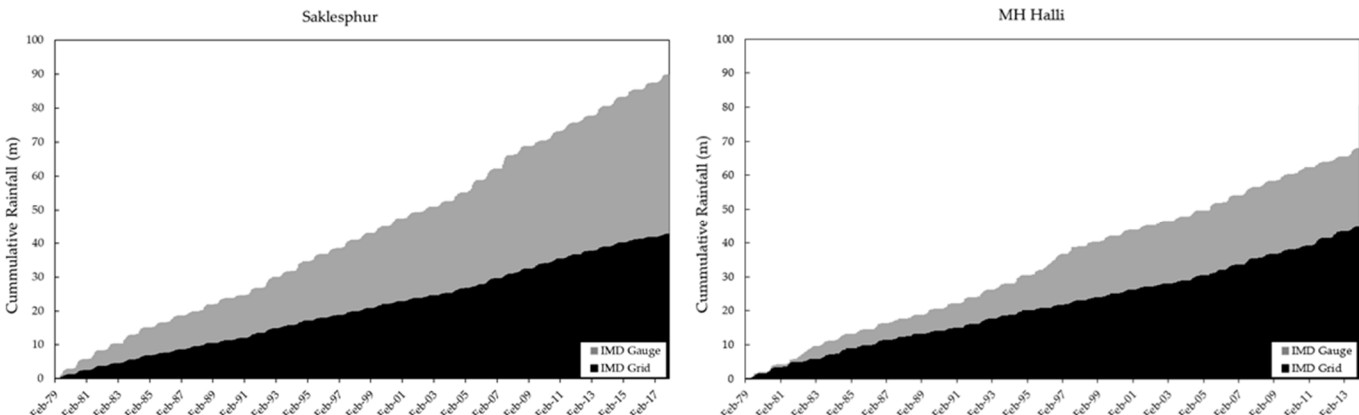

**Figure 8.** The cumulative rainfall (m) from available IMD gauge and gridded sources across Saklesphur and MH Halli sub-catchments (Figure 2) in the headwaters of the Cauvery from 1979 until 2017 and 1979 until 2013 respectively.

Within the model, the reservoir outflow parameters were adjusted within the full range of possible values and combinations to provide the best possible fit to the daily observed outflow data. The temporal signal of the Mettur Dam outflow is noticeable through all the downstream gauges (Urachikottai and Kodumodi) to Musiri. Figure 9a illustrates the ability of the model to better capture the temporal trend of the observed streamflow upstream of Mettur. However, the model was unable to capture the intra- and inter-annual reservoir operations from the Mettur Dam, and thus does not fully represent

the timing of the observed streamflow at Urachikottai downstream of the dam (Figure 9b). The GWAVA reservoir outflow routine is determined by the user-set parameters, as well as the long-term average inflow. The observed reservoir outflows appear to be sporadic and have very little correlation to the reservoir inflow, and thus the reservoir equation within GWAVA does not represent sporadic outflows well.

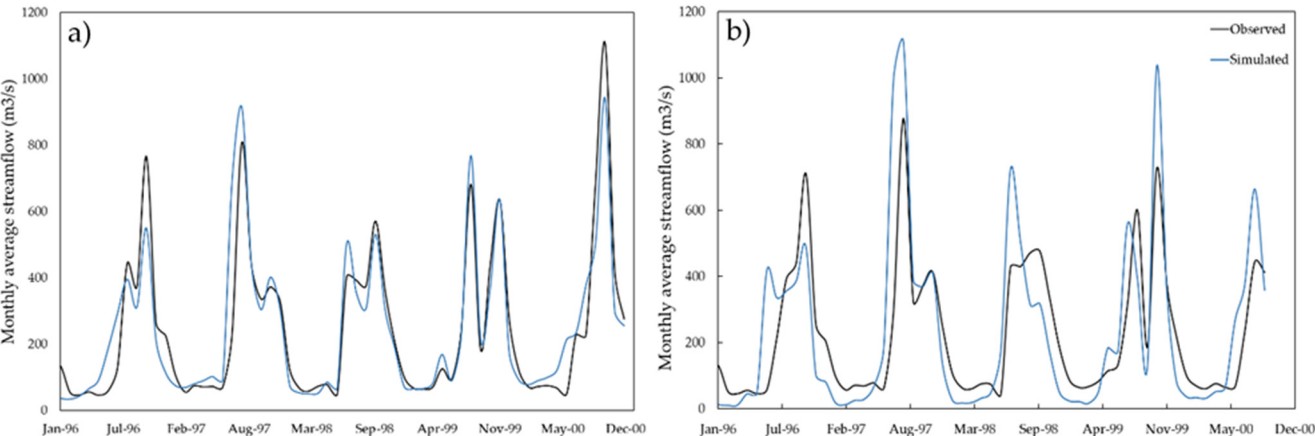

**Figure 9.** The simulated and observed monthly average streamflow from 1996–2000 at (**a**) Bilingudulu gauging station (k-Figure 2), upstream of the Mettur Dam and (**b**) Urachikottai gauging station (l- Figure 2), downstream of the Mettur Dam.

The inclusion of the interventions improves the model performance (presented here as Kling-Gupta Efficiency- KGE). During the calibration and validation, the interventions improved the model performance in nine and seven sub-catchments, respectively, but did not affect the calibration or validation in three and five sub-catchments (Table A2 in Appendix B). The lower KGE in the validation period indicates that the better fitting calibration results could have been obtained due to overfitting of parameters during the calibration process, model equifinality [74], and where the model is not capturing the catchment or reservoir processes correctly. The lack of reliable data pertaining to the Tamil Nadu region and the automatic calibration and conceptual nature of the model could be taking into account processes that are not included in the model structure through the existing model parameterization.

Following calibration and validation of the model, streamflow, quick overland flow, sub-surface flow (water flowing to the stream through the soil profile), base flow (water flowing to the stream from the aquifer), groundwater levels, reservoir storage levels at Mettur Dam, and evaporation for five scenarios were simulated.

### 3.2. Sensitivity Analysis of Interventions within GWAVA

A sensitivity analysis was conducted by altering the density and dimension of the conceptualised interventions within GWAVA. The scenario list of the varying densities and dimension utilised in this analysis can be found in Table A1 in Appendix A. The results of the conceptual tank, check dam, and farm bund sensitivity analyses are presented in Figures A1–A3, respectively in Appendix A.

#### 3.2.1. Tanks

The $Q_{10}$ flow is decreased with the increase in the density and depth of the tanks conceptualised in GWAVA. The $Q_{10}$ is affected by both the increase in depth and density. The mean decreases when the density of tanks is 25 $m^3$/ha, while the mean simulated streamflow increased when increasing density of tanks from 75 to 200 $m^3$/ha. The mean flow is insignificantly affected by the increase in density or depth.

The $Q_{90}$ flow is increased with increasing both the depth and density. As the tanks are deeper than the check dams, the evaporative potential from the surface is less and thus the recharge exceeds the evaporation and is able to contribute to the base flow. The $Q_{10}$ flows

are reduced due to the structural hinderance within the stream channel. When considering the 25 $m^3$/ha density, the $Q_{10}$ is reduced by a greater percentage compared to the increase in $Q_{90}$, thus resulting in a reduction in the mean flow.

### 3.2.2. Check Dams

The $Q_{10}$ flow is decreased with the increase in the density and the dimensions of the check dams conceptualised in GWAVA. The $Q_{10}$ is more sensitive to the increase in dimensions rather than the increase in density. The mean and $Q_{90}$ simulated streamflow is decreased when the density of tanks is 25 $m^3$/ha, while the mean simulated streamflow is increased when increasing density of check dams from 75 to 200 $m^3$/ha. The mean and $Q_{90}$ flow is not highly sensitive to the change in dimensions of the check dams. When the density of conceptualised check dams is less than 25 $m^3$/ha, the rate of evaporation is greater (small volume of stationary water storage) than the rate of recharge and the compounded water is evaporated before it is able to recharge and contribute to the base flow. Once the density of interventions has exceeded 25 $m^3$/ha, the rate of recharge is greater than the rate of evaporation (the volume to surface area ratio is smaller), and thus more water within the check dam is able to recharge and contribute to the base flow component.

### 3.2.3. Farm Bunds

The farm bunds did not have a significant impact on the flows. The changes between the baseline and the four scenarios were less than 1%. Increasing the depth of the farm bunds increased the flows; however, at the spatially scale of GWAVA, this is deemed insignificant, and thus it is concluded that the bunds do not have a significant effect on the flow. Although farm bunds are not constructed on the stream and do not significantly affect the streamflow, they alter other components of the water balance within the basin. The open water surface increases the evaporative potential across the field, and the pooling water increases the soil water for the period following a rainfall event.

### 3.3. Effect of Interventions in the Cauvery

In this section, all observations are drawn from model simulations (i.e., simulated streamflow, base flow, evaporation, and groundwater level). The effects of interventions on simulated streamflow across the modelling period (1986–2005) were evaluated using the mean flow ($\overline{Q}$), the flow exceeded 90% of the time ($Q_{90}$, quantification of low flows) and the flow exceeded 10% of the time ($Q_{10}$, representation of high flows). Additionally, the effects of the interventions on the simulated streamflow and evaporation, in a wet (2005), normal year (1998), and dry (2002), year at the catchment outlet of S1 and S2 and Musiri, were investigated. These years were chosen by considering the lowest, highest, and mean total annual precipitation across the catchments (Table 2).

**Table 2.** The total annual precipitation and the reduction in flows days with the inclusion of interventions for the selected catchments S1, S2 (Figure 2), and Musiri (Figure 2) for wet, dry, and normal year.

| Sub-Catchment | Total Annual Precipitation (mm) | | | Reduction in Flows Days with the Inclusion of Interventions | | |
|---|---|---|---|---|---|---|
| | Normal Year (1998) | Dry Year (2002) | Wet Year (2005) | Normal Year (1998) | Dry Year (2002) | Wet Year (2005) |
| S1 | 507 | 382 | 668 | 14 | 25 | 3 |
| S2 | 1874 | 656 | 2085 | 2 | 4 | 3 |
| Musiri | 1341 | 685 | 1413 | 0 | 0 | 0 |

In the non-perennial catchment (S1, Table 1 and Figure 2), the surface flow is the dominant component of the simulated streamflow (Figure 10). The simulated streamflow ($Q_{10}$, $\overline{Q}$ and $Q_{90}$) is reduced with the inclusion of interventions. However, it is the high flows, $Q_{10}$, that are more significantly reduced (Figure 11). The interventions have a greater impact on the simulated streamflow in S1 than S2. The simulated streamflow is reduced to the largest extent in the normal year (~10%, Figure 11a). The stormflow is intercepted by the intervention and, thus, reduces the simulated streamflow in the wet season ($Q_{10}$,

Figure 11b). The dry season flows ($Q_{90}$, Figure 11b) are reduced as any subsurface lateral flow (from the soil store) entering the stream is impounded by the intervention. The stormflow component is larger than the subsurface lateral flow and base flow components in this catchment and, thus, the simulated streamflow is affected to a greater extent in the wet season. The non-perennial streams dry out earlier with the inclusion of interventions (Figure 12 and Table 2). The total evaporation across the sub-catchment is increased with the inclusion of interventions with the greatest increase occurring in the wet year (Figure 11a) as water is present in the interventions for a greater length of time. In this catchment, the water table is increased in the wet season with the inclusion of interventions (Figure 13). Despite the increase in simulated recharge, the water table does not reach a level where the water in the groundwater will contribute significantly to simulated base flow.

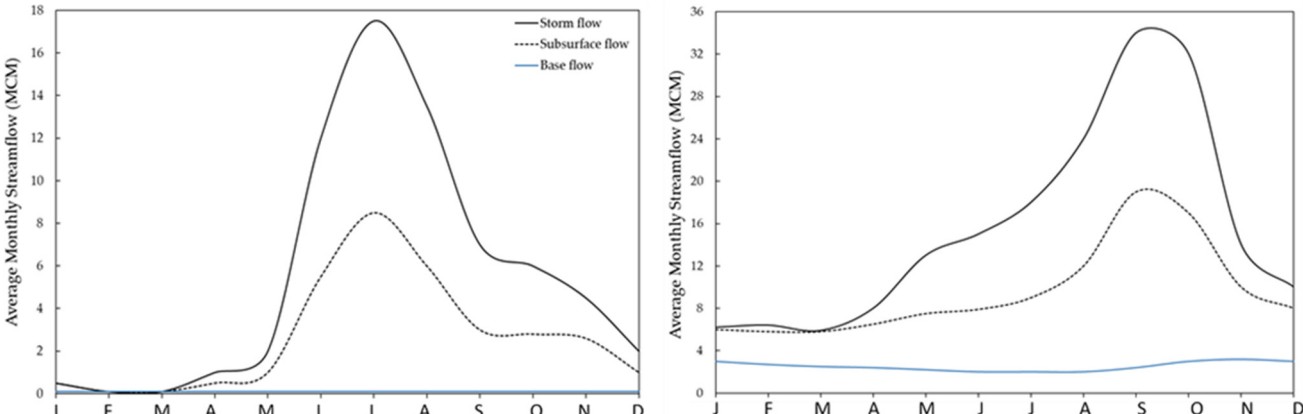

**Figure 10.** The mean monthly simulated separation hydrograph for S1 (**left**) and S2 (**right**) (Table 1 and Figure 2) from 1986 until 2005 with all interventions included.

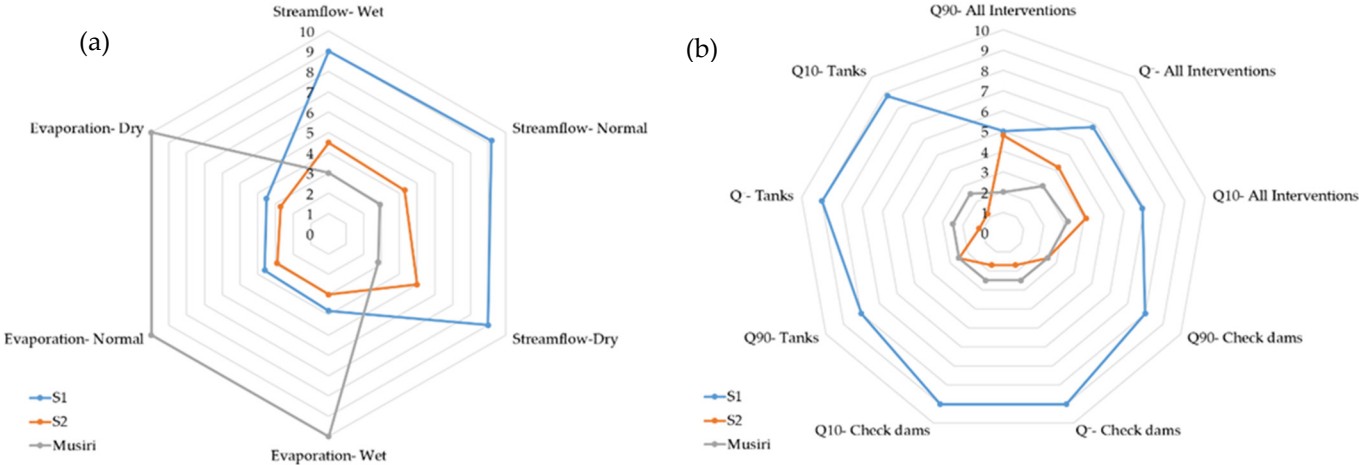

**Figure 11.** (**a**) The percent reduction in total annual simulated streamflow and increase in total annual simulated evaporation (%) with the inclusion of interventions for S1, S2, and at Musiri in wet (2005), normal (1998), and dry (2002) years. (**b**) The effect of all the interventions (tanks, check dams and bunds); check dams only and tanks only on high flows ($Q_{10}$); low flows ($Q_{90}$); and mean flows ($\overline{Q}$) flows across S1, S2, and at Musiri (Table 1, Figure 2).

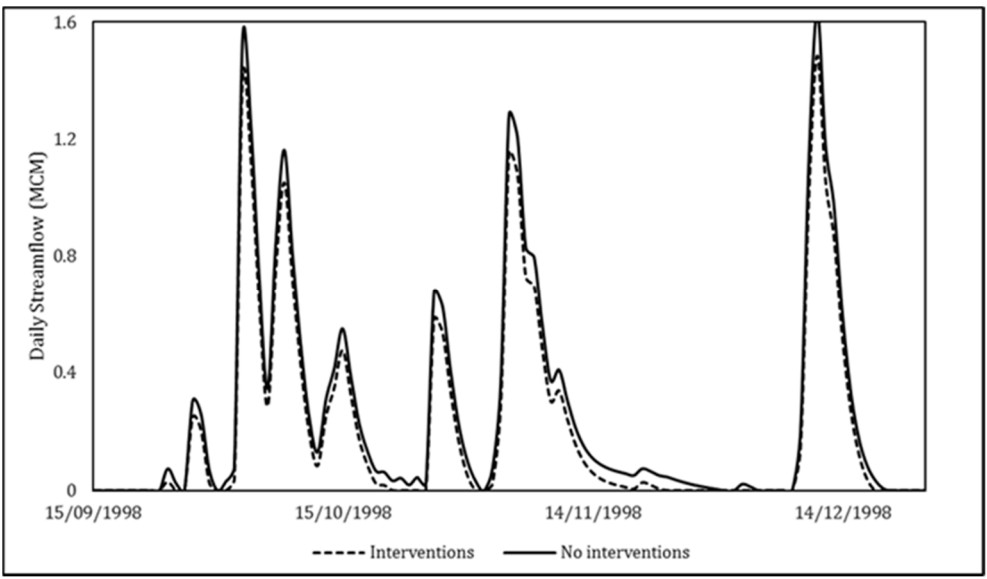

**Figure 12.** An example of the simulated streamflow in sub-catchment S1 (Table 1, Figure 2) with interventions and without interventions through the period of September 1998 until December 1998 (normal year).

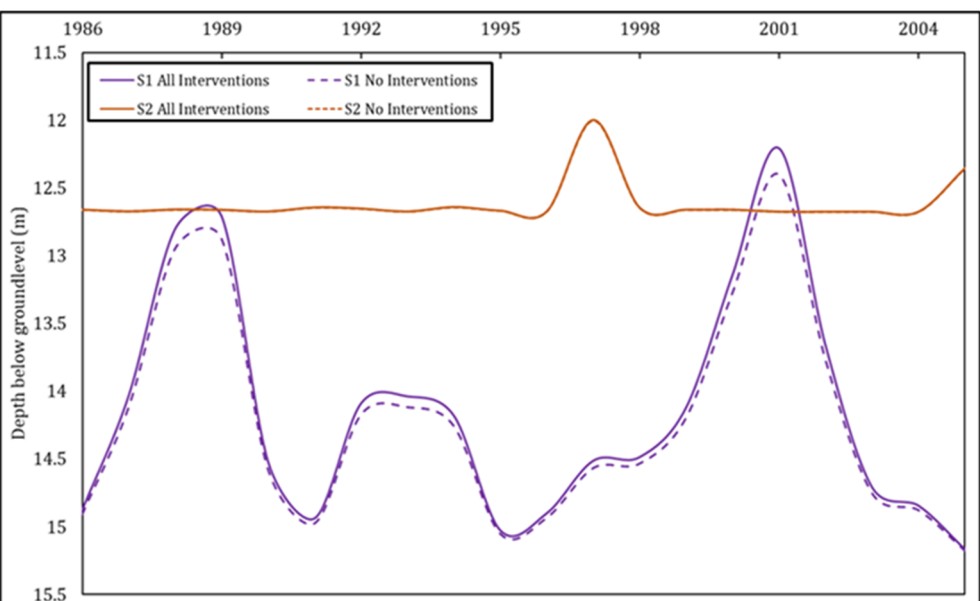

**Figure 13.** Monthly mean groundwater level below ground surface for S1 and S2 (Table 1 and Figure 2) with and without the inclusion of interventions. There is no change to the aquifer level in S2—this is illustrated by the S2 no interventions line (orange dash), which falls directly on S2 all interventions line (orange line).

In the perennial catchment (S2, Table 1 and Figure 2), the stormflow is dominant in the wet season but the subsurface flow and base flow is dominant in the dry season (Figure 10). The simulated streamflow ($Q_{10}$, $\overline{Q}$ and $Q_{90}$) is reduced, and the $Q_{90}$ is more significantly reduced with the inclusion of all interventions (Figure 11b). The interventions have a similar effect on simulated streamflow in the dry and wet years (~5%, Figure 11a). In the wet season, the simulated streamflow is reduced due to the in-situ impoundment, and the low flows are maintained but reduced in the dry season. In the dry season, simulated streamflow is reduced because the base flow and any subsurface lateral flows (from the soil store) entering the stream are impounded by the intervention. The impounded water is

subject to both evaporation and recharge. The total evaporation across the sub-catchment increased with the inclusion of interventions with the greatest increase occurring in the wet and normal years (Figure 11a) as there is water in the interventions for a greater length of time. In this catchment, the groundwater level is minimally affected by the inclusion of the interventions (Figure 13). The water table is above the level at which the groundwater will flow as base flow. Base flow will continue to occur but simulated streamflow will be reduced in the dry season as any simulated streamflow produced by the base flow above the intervention will be impounded.

The responses of the S1 and S2 and Musiri exhibit opposite responses. The change in simulated streamflow in S1 and S2 is greater than the change in evaporation while at Musiri, the change in evaporation is greater than the change in simulated streamflow. At Musiri, the simulated streamflow is dominated by the Mettur Dam releases (Figure 2). The interventions do not have a significant effect on the simulated Mettur Dam release flows. The minimal reduction in mean simulated streamflow (~3%) seen at Musiri can be considered as the consequence of the interventions in the tributaries that join the main Cauvery channel downstream of Mettur Dam. However, on analysis of the effect of interventions on the inflow into Mettur Dam, it was found that the interventions reduced the mean simulated streamflow ($\overline{Q}$) by ~6% and the simulated streamflow in the wet season ($Q_{10}$) was reduced by ~26%. This demonstrates that the large reservoir has the ability to nullify the impact of the interventions, however, their effect can be seen in the reduction of simulated streamflow entering the reservoir. In this unique case, the effect of the interventions on the Mettur Dam inflow is more representative of the effects of interventions at a basin-scale than those shown at Musiri and corresponds more correctly with the increase in total evaporation across the basin with the inclusion of interventions of ~10% (Figure 11a).

The majority of the flow into the Cauvery Basin is contributed to by sub-catchments [75] along the western boundary (Figure A4 in Appendix D). However, most of the interventions are constructed in semi-arid regions. The simulated $Q_{90}$ flow in these humid catchments is affected more by the interventions than in the semi-arid sub-catchments on the eastern boundary (Figure A4 in Appendix D). Conversely, there is a greater effect of the interventions on the simulated $Q_{10}$ flow in the semi-arid sub-catchments (Figure A4 in Appendix D). The effect on the $Q_{10}$ flows is greater in the semi-arid sub-catchments because the monsoonal streamflow is required to fill these structures before they begin to spill. In the humid catchments, the interventions do not have a great effect on the $Q_{10}$ flow as it is likely to be the presence of water within these structures before the monsoon and the intervention immediately spills. Although the percent change in $Q_{10}$ flows in the semi-arid sub-catchments is higher, the volume of water impeded in these structures may be greater in the humid sub-catchments. The $Q_{90}$ flow is impacted more severely in the humid catchments as these streams are fed during the dry season through base flow, whereas in the semi-arid sub-catchments, the streams frequently run dry with or without interventions. The implementation of interventions in these sub-catchments stores water further up in the basin and essentially impedes the downstream flow. The reallocation of water by these structures limits water, especially the monsoon flows, from entering the ocean unused, and provides an inexpensive means of decentralised water management. Although these structures allow available water to be utilised throughout the basin, there are subsequent implications for users and environmental flows downstream when low flows are reduced. A summary of the total changes in precipitation, simulated streamflow (with and without interventions), simulated evaporation (with and without interventions), and aquifer levels (with and without interventions) can be found in Table A4 in Appendix D.

## 4. Discussion

The model calibration was acceptable in the upper reaches of the basin, but the model fit was weaker downstream of the Mettur Dam (Figure 2). The inclusion of interventions improves the model performance. It provides a better account of the surface storage

within the basin and better estimation of the time of concentration in the sub-catchments without major reservoirs. The farm bunds were found to have little effect on the simulated streamflow, as the high water demands of the rainfed crops cause the infiltrated water from the farm bunds to be transpired quickly and there to be little difference in the water converted to base flow or groundwater recharge with or without the bunds. Assuming the relationship between the area of farm bunds and the area of rainfed cropland determined for Karnataka holds into Tamil Nadu, the majority of the bunds were located within the lower regions of the basin where there is a greater area of rainfed cropland. It is difficult to distinguish the exact effects of the farm bunds in these regions as the river system is heavily dominated by the Mettur Dam outflows. Farm bunding proves to be an effective method of increasing soil water in areas of rain-fed agriculture and mitigating water allocation challenges, and aggressive groundwater pumping associated with intensive irrigation, without signicantly effecting the basin hydrology. However, conceptually, the model fills the farm bunds followed by the tanks and then the check dams. In the simulations with all the interventions included, this limits the water available for filling the tanks and check dams. Although individually the bunds have little effect on the simulated streamflow, when simulated with the tanks and check dams, the reduction in water available to fill tanks and check dams is reflected in the lower impact on the simulated streamflow. Individually, the tanks and check dams have a similar effect on the simulated streamflow (Figure 11b).

A significant challenge in large-scale hydrological modelling is quantifying and managing the uncertainty in climate forcing and evaluation data. Uncertainty can arise from observation gauge density, spatial and temporal interpolation methods, and general measurement errors. The Western Ghats region in the NW of the basin is a known area of uncertainty with the IMD precipitation data [73]. Each 0.5-degree grid cell contains numerous terrain and gradient increments, and the grid cells fall over the basin boundary. This results in an inaccurate representation of the distribution and total rainfall, as well as the distribution of minimum and maximum temperature in this region of the basin [76]. There is a significant source of uncertainty as this region acts as the headwaters for the larger Cauvery Basin (Figure 8). At some gauging points in the basin, there is low confidence in the observed streamflow data [77]. Eye-witness accounts and some literature [78] report the drying out of streams in the dry season, which is not reflected in the observed data. Additionally, in reality, rivers downstream of significant urban areas (Arkavathy downstream of Bangalore and Eluthunimangalam downstream of Coimbatore and Tiruppur) are fed by a perennial stream of sewage [78]. The model represents return flows from domestic demand, but this may be underestimated compared to the volume of effluent being actually released into these rivers. The analysis of the precipitation and the observed streamflow, used within this study, showed temporal discrepancies. The temporal difference between rainfall events and the hydrograph peak did not show a systematic error or a consistent lag time.

The scale of this study (0.125 degree) required the aggregation of the surface area of each type of intervention in each cell. The simplification in the conceptualisation of the interventions is a cause of uncertainty in this study [79]. The aggregation of the interventions into one composite tank, check dam, and farm bund within the cell, skews the surface area to capacity ratio. As intervention data were limited to surface area, if one calculates the intervention capacity from the combined surface area, the capacity is greater than calculating the capacity of each individual intervention and aggregating the capacity. This causes the holding capacity of the conceptual interventions in each cell to be greater than in reality. Subsequently, the larger conceptual intervention will not fill or spill as frequently as many smaller interventions, and thus the estimation of the effect on simulated streamflow of all the interventions is uncertain. Additionally, the evaporation could be underestimated as a larger waterbody requires increased energy for evaporation and has a larger lag time (due to heat storage) than a smaller one. This may also lead to the individual smaller interventions being subjected to more evaporative losses than these estimated in the model using the larger conceptual intervention. Conversely, the model structure allocates water to the evaporative component first, and thus, the evaporative processes are favoured

in times of water stress. This could, along with the use of the Hargreaves evaporation estimation method, additionally be one of the fundamental reasons for the systematic underestimation of simulated streamflow across the basin [80]. The aggregation of the cascading tank systems into one large tank, and numerous check dams into one large check dam, results in the true effects of the cascading system not being represented within the model [10]. Numerous tanks and check dams on a river network can cause streamflow in the river and the subsurface and base flow emerging into the stream to be obstructed by the downstream check dams.

Due to a lack of data, the process of quantifying the distribution of the interventions across the basin relies upon many assumptions and, thus, generates significant uncertainty. The accuracy of the Structural Investment Report is unknown and the assumption of a fixed cost per structure/hectare across Karnataka is unlikely to be accurate. Similarly, assuming that the systems and behavioural patterns (agricultural practices, usage of infrastructure, etc.) in the state of Karnataka and Tamil Nadu are identical is also unlikely. However, due to data scarcity and lack of evidence to validate these assumptions, a pragmatic approach was used to allow the inclusion of small-scale interventions in a large-scale hydrological model.

In the absence of data to formally validate a new concept introduced into a hydrological model, it is important to measure results against exisiting literature. Despite the uncertainty and pilot nature of this study, the trends identified within the Cauvery Basin are in line with the findings from Garg et al. (2012) [13]. Garg et al. [13] altered a number of parameters within SWAT (surface runoff, water holding capacity, available soil water, groundwater recharge, and curve number) to reflect the potential influence of the check dams and farm bunds in the basin, and found that the interventions have a slightly greater effect on the simulated streamflow in wetter years. The study also found that the largest portion of the water balance is the evaporatative component and the evaporative losses increased with the inclusion of the interventions. This is in agreement with this study (Figure 11b). Garg et al. (2012) [13] found that check dams reduced the annual simulated streamflow at the basin outlet of the Kothapally catchment by 9%. This corresponds with the GWAVA simulation, which estimated ~9% reduction in simulated streamflow (Figure 11a) in S1 of similar MAP, soil type, and land use. In contrast, the groundwater recharge from the individual interventions was significant in the Garg et al. (2012) [13] study.

There is also agreement between the results of S2 and the work of Xu et al. (2013) [11], in which they concluded that check dams reduce the total runoff in the rainy season (15%, Figure 13). Xu et al. (2013) [11] did not specifically include the characteristics of the interventions, but rather attributed the difference between a period of observed and simulated streamflow as the effect of the interventions. The decrease in mean annual streamflow (14%) estimated by Xu et al. (2013) [11] and attributed to the effect of check dams does correlate to the decrease in mean annual streamflow of S2 as a result of check dams (15%, Figure 10). Sub-catchment S2 has a similar MAP and type of vegetation as the catchment studied by Xu et al. (2013) [11]. This is an important correlation, with two different approaches yielding similar results. In the absence of data pertaining to pre- and post-intervention construction, conceptulising the structures within a model could provide accuarate estimations of their influence.

The decrease in simulated streamflow by GWAVA in S1 and S2 due to tanks was 4% and 5%, respectively. These results differ significantly from those of Van Meter et al. (2015) [10], where the simulated streamflow was found to decrease by 75% from a single cascading tank system in a catchment with an MAP of 850 mm in Tamil Nadu. GWAVA conceptualises the tank systems within a cell into one large hypothetical tank, and thus does not capture the cascading characteristics of the tank systems. This could explain the difference in the observed streamflow reduction; alternatively, the tank system investigated by Van Meter et al. (2015) [10] could be atypical.

GWAVA may not capture the sensitivity of hydrological fluxes at a local-scale as a well as some catchment-scale models [10,11,13]. However, yielding similar results to that

of published small-scale studies provides a good starting point for further refinement of the conceptualisation within large-scale hydrological models.

## 5. Conclusions

The modified version of GWAVA provided a valuable tool to investigate the effects of interventions at a sub-catchment and basin scale.

The main conclusions from this study are:

- Conceptualised interventions play an important part in the allocation and better representation of simulated surface water within the basin.
- The effect of the conceptualised interventions within GWAVA is dependent on the hydrogeology of the modelled sub-catchment, as well as the simulated groundwater level.
- The influence of the interventions is greater on the simulated streamflow in the wet years and on estimated evaporation in the dry years.
- Farm bunds provide an effective method for reducing the pressure of canal irrigation and groundwater pumping in agricultural areas.

The results of this study corresponded well with existing literature from small-scale studies. However, at the sub-catchment and basin scale, groundwater levels appear less impacted than in the cited literature or indigenous knowledge surrounding the use of interventions for water security at a local scale, suggesting further investigation is required. This study incorporated stakeholder and expert knowledge, as well as published literature information in the conceptualisation of the interventions within the model. New and creative approaches had to be utilised where data gaps existed to model the effects of interventions at the basin scale. The approach outlined in this study can be applied in different model applications in regions where interventions are prominent, if the source code is available for adaption. This study had to rely on a pragmatic approach, and as a consequence, many assumptions were made. It does, however, provide a step forward in the conceptualisation, quantification, and implication of small-scale storage interventions at the basin scale.

**Author Contributions:** Conceptualisation, R.H., P.S.W., N.R., M.S., V.S., K.K.G. and V.J.D.K.; methodology, R.H., H.E.B. and P.S.W.; software, R.H., H.E.B. and V.J.D.K.; validation, R.H. and H.E.B., formal analysis, R.H.; investigation, R.H. and H.E.B.; data curation, P.S.W., K.K.G., R.H., writing—original draft preparation, R.H.; writing—review and editing: V.J.D.K., H.E.B., V.S., N.R., H.G.R., H.A.H.-C.; supervision, V.J.D.K., H.A.H.-C. and H.G.R.; project administration, H.G.R., H.A.H.-C. and V.J.D.K. All authors have read and agreed to the published version of the manuscript.

**Funding:** The research underlying this paper was carried out under the UPSCAPE project of the Newton-Bhabha program "Sustaining Water Resources for Food, Energy and Ecosystem Services", funded by the UK Natural Environment Research Council (NERC-UKRI) and the India Ministry of Earth Sciences (MoES), grant number: NE/N016491/1. UK Centre for Ecology and Hydrology (UKCEH) publish with the permission of the Director of UKCEH. The views and opinions expressed in this paper are those of the authors alone.

**Institutional Review Board Statement:** Not applicable.

**Informed Consent Statement:** Not applicable.

**Data Availability Statement:** Datasets utilised for this research are available in these in-text data citation references: Pai et al. (2014) [73], Central Water Commission (1987) [53], Wable, et al. (2019) [62], NASA JPL (2013) [81], Fischer et al. (2008) [82], Roy et al. (2008) [83] and Robinson et al. (2014) [84]. The data contained within the Structural Investment Report of Karnataka and any other data presented in this study are available on request from the corresponding author.

**Conflicts of Interest:** The authors declare no conflict of interest.

# Appendix A

**Table A1.** Description of scenarios utilised in the sensitivity.

| Scenario | Depth (m) | Width (m) | Intervention Density (m³/ha) | Scenario | Depth (m) | Width (m) | Intervention Density (m³/ha) |
|---|---|---|---|---|---|---|---|
| Tanks | | | | | | | |
| T1 | 3 | n/a | 25 | T7 | 5 | n/a | 125 |
| T2 | 3 | n/a | 75 | T8 | 5 | n/a | 200 |
| T3 | 3 | n/a | 125 | T9 | 10 | n/a | 25 |
| T4 | 3 | n/a | 200 | T10 | 10 | n/a | 75 |
| T5 | 5 | n/a | 25 | T11 | 10 | n/a | 125 |
| T6 | 5 | n/a | 75 | T12 | 10 | n/a | 200 |
| Check Dams | | | | | | | |
| C1 | 1 | 7 | 25 | C7 | 1.5 | 10 | 125 |
| C2 | 1 | 7 | 75 | C8 | 1.5 | 10 | 200 |
| C3 | 1 | 7 | 125 | C9 | 2 | 15 | 25 |
| C4 | 1 | 7 | 200 | C10 | 2 | 15 | 75 |
| C5 | 1.5 | 10 | 25 | C11 | 2 | 15 | 125 |
| C6 | 1.5 | 10 | 75 | C12 | 2 | 14 | 200 |
| Farm Bunds | | | | | | | |
| B1 | 0.03 | n/a | 25 | B5 | 0.06 | n/a | 25 |
| B2 | 0.03 | n/a | 75 | B6 | 0.06 | n/a | 75 |
| B3 | 0.03 | n/a | 125 | B7 | 0.06 | n/a | 125 |
| B4 | 0.03 | n/a | 200 | B8 | 0.06 | n/a | 200 |

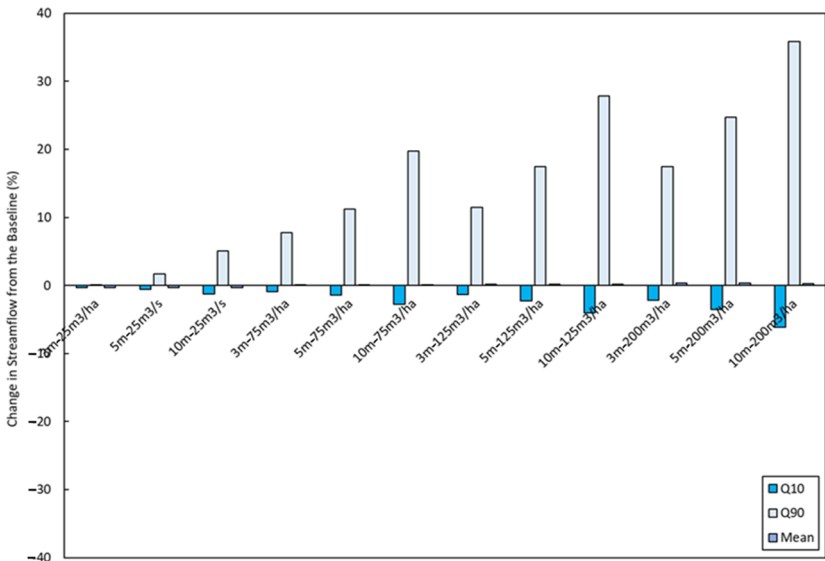

**Figure A1.** The change in the $Q_{10}$. $Q_{90}$ and mean simulated streamflow from the baseline with the inclusion of tanks of various density and depths (Table A1).

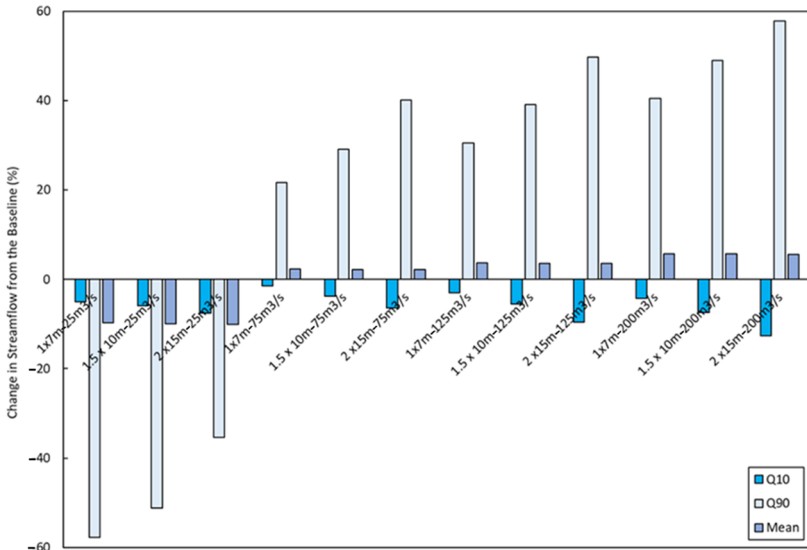

**Figure A2.** The change in the $Q_{10}$. $Q_{90}$ and mean simulated streamflow from the baseline with the inclusion of check dams of various density and dimensions (Table A1).

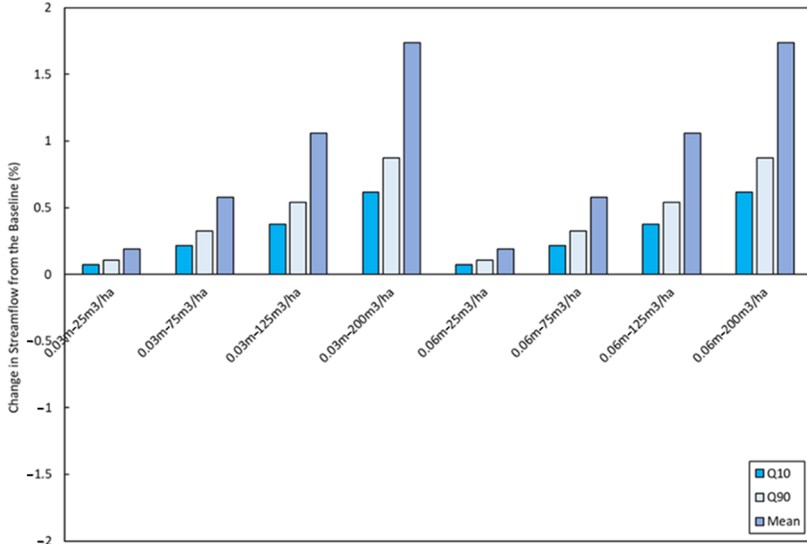

**Figure A3.** The change in the $Q_{10}$. $Q_{90}$ and mean simulated streamflow from the baseline with the inclusion of farm bunds of various density and dimensions (Table A1).

## Appendix B

**Table A2.** Calibration and Validation monthly Kling-Gupta Efficiency (KGE) values when GWAVA was calibrated with and without the inclusions of interventions. The green shading represents a better performance, the grey shading represents the same performance, and the orange shading represents a poorer performance with the inclusion of interventions.

| | Sub-Catchment | Calibration | | | Validation | | |
|---|---|---|---|---|---|---|---|
| | | **KGE without Interventions** | **KGE with Interventions** | **Period** | **KGE without Interventions** | **KGE with Interventions** | **Period** |
| a | Saklesphur | 0.53 | 0.66 | 2006–2010 | 0.37 | 0.38 | 2010–2013 |
| b | Thimmanahali | 0.71 | 0.71 | 2005–2009 | 0.72 | 0.68 | 2010–2013 |
| c | KMVadi | 0.25 | 0.24 | 1991–2000 | 0.16 | 0.16 | 2001–2011 |
| d | Kudige | 0.48 | 0.59 | 1990–2000 | 0.55 | 0.59 | 2012–2014 |
| e | Munthankera | 0.73 | 0.82 | 1990–2000 | 0.66 | 0.70 | 2001–2011 |
| f | Tbekuppe | 0.41 | 0.32 | 1980–1990 | −1.28 | −1.27 | 2001–2003 |
| g | TKHali | 0.52 | 0.71 | 1990–2000 | 0.69 | 0.67 | 2001–2008 |
| h | T Narasupiar | 0.60 | 0.68 | 1988–1998 | 0.25 | 0.30 | 1999–2002 |
| i | Kollegal | 0.56 | 0.58 | 2008–2011 | 0.50 | 0.54 | 2012–2013 |
| j | Bilingudulu | 0.74 | 0.74 | 1990–2000 | 0.61 | 0.60 | 2001–2011 |
| k | Urachikottai | 0.34 | 0.74 | 1990–2000 | 0.49 | 0.51 | 2001–2008 |
| l | Kodumodi | 0.25 | 0.62 | 1990–2000 | 0.24 | 0.30 | 2005–2010 |
| o | Musiri | 0.33 | 0.65 | 1990–2000 | 0.43 | 0.44 | 2006–2010 |
| m | Thengumarahada | 0.57 | 0.50 | 1990–2000 | 0.39 | 0.44 | 2001–2008 |

# Appendix C

**Table A3.** Input data utilised in the GWAVA model setup.

| Input Data | Spatial Resolution | Temporal Resolution | Time Period | Source |
|---|---|---|---|---|
| Precipitation | 0.25 degree | Daily | 1951–2017 | Indian Meteorological Department [73] |
| Maximum Temperature | 0.25 degree | Daily | 1951–2016 | Indian Meteorological Department [73] |
| Minimum Temperature | 0.25 degree | Daily | 1951–2016 | Indian Meteorological Department [73] |
| Open Water Evaporation | India | Monthly | 1959–1968 | Central Water Commission, Basin Planning & Management Organisation [53] |
| Streamflow gauged data | Cauvery Basin | Daily | 1971–2014 | India-WRIS |
| Reservoir inflow and outflow data | Cauvery Basin | Monthly | 1974–2017 | India-WRIS |
| Water transfers | Cauvery Basin | | | Ashoka Trust for Research in Ecology and the Environment (ATREE) |
| Tanks | Cauvery Basin | | 2019 | Waterbodies dataset [71] |
| Check dams | Karnataka (District) | | 2006–2012 | Structural Investment Report, Watershed Development Department |
| Farm bunds | Karnataka (District) | | 2006–2012 | Structural Investment Report, Watershed Development Department |
| Elevation | 0.003 degree | | 2000 | NASA Shuttle Radar Mission Global 1 arc second V003 [81] |
| Soil type | 0.008 degree | | 1971–1981 | Harmonized World Soil Database v1.2 [82] |
| Land Cover Land Use | 0.001 degree | | 2005 | Decadal land use and land cover across India 2005 [83] |
| Crops | Cauvery Basin (Taluk *) | | 2000 | National Remote Sensing Centre (NRSC) |
| Total Population | Cauvery Basin (Village) | | 2011 | Indian Decadal Census |
| Rural Population | Cauvery Basin (Village) | | 2011 | Indian Decadal Census |
| Livestock | 0.05 degree | | 2005 | CGIR Livestock of the World v2 [84] |

* Taluk-a subdivision of a district consisting of a group of several villages organized for revenue purposes.

**Appendix D**

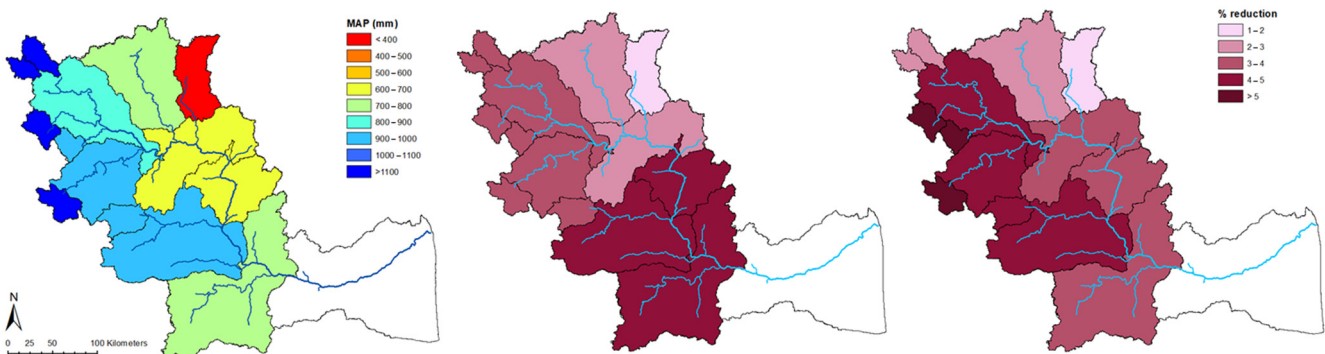

**Figure A4.** (**Left**): the mean annual precipitation for each calibration catchment over the period 1986–2005; middle: the percentage reduction in $Q_{10}$ flow for each calibration catchment over the period 1986–2005; (**right**): the percentage reduction on $Q_{90}$ flow for each calibration catchment over the period 1986–2005.

**Table A4.** The total annual (for 1998, 2002 and 2005) precipitation (P), simulated streamflow (Q), simulated total evaporation (ET) and average annual aquifer level (Aq) below ground level with and without interventions. The change (Δ) with the inclusion of interventions is presented as a percentage change.

| | Year | P (mm) | Q-int (mm) | Q-no int (mm) | ΔQ (%) | ET-int (mm) | ET-no int (mm) | ΔET (%) | Aq-int (m) | Aq-no int (m) | ΔAq (m) |
|---|---|---|---|---|---|---|---|---|---|---|---|
| | 1998 | 507 | 118 | 130 | −9.4 | 624 | 602 | 3.4 | 14.02 | 14.42 | 2.7 |
| S1 | 2002 | 382 | 44 | 48 | −8.8 | 436 | 421 | 3.4 | 12.46 | 12.86 | 3.1 |
| | 2005 | 668 | 68 | 75 | −8.6 | 735 | 708 | 3.6 | 14.52 | 14.92 | 2.6 |
| | 1998 | 1874 | 977 | 1020 | −4.2 | 1527 | 1485 | 2.7 | 12.61 | 12.61 | 0 |
| S2 | 2002 | 656 | 669 | 704 | −5 | 489 | 477 | 2.5 | 12.64 | 12.64 | 0 |
| | 2005 | 2085 | 802 | 840 | −4.5 | 1301 | 1262 | 3 | 12.32 | 12.32 | 0 |
| | 1998 | 1341 | 325 | 334 | −2.6 | 1030 | 928 | 9.9 | 8.97 | 9 | 0.2 |
| Outlet | 2002 | 685 | 130 | 134 | −2.6 | 521 | 469 | 9.9 | 8.97 | 8.99 | 0.3 |
| | 2005 | 1413 | 432 | 446 | −3.1 | 1067 | 962 | 9.8 | 8.95 | 8.96 | 0.1 |

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
