# Peer review of "Modelling Small-Scale Storage Interventions in Semi-Arid India at the Basin Scale"

_sustainability, doi:10.3390/su13116129_

Round 1

Reviewer 1 Report

Dear Authors,

the paper presents model research of storage interventions on the example of basin-scale in India. The topic can be considered interesting and original, especially in terms of using remote sensing techniques and GWAVA model in water (and sediment) management in catchment (and basin) scale. Results are interesting in the research of experimental measurements in terms of artificial reservoirs functioning. I made comments on selected lines or parts of the manuscript. And I hope they will help the authors to improve the flow of the manuscript.

I suggest changing the name “tank” to “reservoir” – terms “tank” in my opinion is not convergent with a type of analyzed structures in terms of use in the literature.

Please introduce the influence of relatively small farm bunds on water balance in the catchment – in ms that information is not clear. I understand that for the functioning of the model water balance is important, but there are not directly connected with the river stream. You also write it in 476 lines.

Usually “check dams” are located in mountain areas and their purpose is “catching material transported while high water flushed from catchment” and usually they are dry-dams (without water storage). Please provide information on why “check dams” that you analyzed in the model are different aims, it means to have water storage and are directly connected with water streams – are an active part of fluvial processes.

I suggest provide more information about the influence of complicated hydrogeology of the study area on the water balance of the catchment.

A general remark, sometimes authors provide too detailed information that is hard to understand the localization of study area: all basin or sub-catchments only, eg. line 411-422.

Please add clear information for a model period, because input data have different scales; results that you present have also different periods – please explain differences, because after reading ms is not clear.

The discussion part contains only 5 references! to the literature source, as a state of knowledge part – it is quite poor – It is a very good discussion of results but almost without the state of knowledge – I suggest add more references to similar processes that you show on result part of ms.

In the conclusion part “main conclusions” are in two sentences (line 709-711) rest of this part is general conclusions – please specify the main conclusions following your results and discussion.

I add some comments on different lines of the document.

L60-96: You wrote about local hydrological conditions, it’s hard to catch all information without a map of the Cauvery Basin – maybe you refer to figure 1 and add names of province/basins to figure, will be easier to read it.

L98: “Remote sensing methods” is too wide – please clarify

L98: “ANN model” you use “ANN” the first time – please add the full name of the model and then in () you can add a shortcut

L100: “SCS-CN” as above

L100: “VIC-MHM” as above

L106: “etc” dot is missing

L107: “etc” as above

L185: Figure one – please add scale bar, north arrow; if on the figure is Cauvery Catchment that is wrong delimitated – why on right (East side) rivers are flowing directly to the Bay of Bengal? Please change the basin border; it's real streamflow or delimitated from DEM?; quality of the figure is poor – please improve it.

L188-189: Please provide in the manuscript (ms) more information about the three reservoirs that you pointed out in figure 1 – I understand that they have a big capacity, etc.

L305: “0.125°” this is “angular measure unit” – please provide “metric unit”; please confirm that in the model you use the metric unit, yes?

L328: the period in ms “1980 until 2013” is not compared with periods in Appendix C – relatively in all gauging stations is very short – extremely 3 years in some places – please provide more explanation about errors in calibration and validation processes in terms of period time.

L332: In ms Appendix link (A, C, B…) are not in sequence – please change it.

L355: “Decadal land use and land cover across India 2005” are available youngest data sources?

L376: On Equation is “m2” but on Figure 4 is “ha” – please unique.

L398: Figure 6 north arrow and scale bar is missing; the legend is poor quality; the range is not clear (dot and blue color)

L405: “Figure 9).” links are not in sequence

L414: “Urachikottai and Kodumodi” I don’t understand where are these gauging stations? Please add localization on figure 1;

L423: Please unique Y-axis on figure 7; legend (blue and orange color) is not compare with figure (gray colors)

L570: Figure 9 is illegible

L577: Figure 10 is illegible

L620: “(2015)” the method of citation is not consistent with the requirements of the publisher (MDPI) – see instructions for authors

L740: “m3/ha” please “3” as the top mark

L759: Map in appendix D are not compare with figure 1 – please explain why? What is a real study area? All catchment (with an outlet to the Bay) or (as Appendix D) or how on Figure 6 (grid)?

Best regards

Author Response

Please see document attached

Reviewer 2 Report

The manuscript would benefit greatly from the help of a good technical editor. The abstract and conclusions sections should be bolstered. There are several sections where the authors need to address: So what? What is the significance and implications? The authors should address the various comments/suggested revisions contained in the technical review.

Author Response

Please see document attached

Round 2

Reviewer 2 Report

See attached review.
